# A Novel Design Nomogram for Optimization of Micro Search Coil Magnetometer for Energy Monitoring in Smart Buildings

**DOI:** 10.3390/mi13081342

**Published:** 2022-08-18

**Authors:** Hadi Tavakkoli, Kui Song, Xu Zhao, Mingzheng Duan, Yi-Kuen Lee

**Affiliations:** 1Department of Mechanical and Aerospace Engineering, Hong Kong University of Science and Technology, Hong Kong, China; 2School of Mechanical Engineering and Mechanics, Xiangtan University, Xiangtan 411105, China

**Keywords:** micro search coil magnetometer, signal-to-noise ratio, energy monitoring

## Abstract

In this paper, a new analytical method to achieve the maximum signal-to-noise ratio (*SNR*) of a micro search coil magnetometer (µSCM) is presented. A planar spiral inductor was utilized to miniaturize conventional bulky search coil magnetometers. First, dimensional analysis was applied to identify three dimensionless parameters for the µSCM’s key performance indices (sensitivity (*S_e_*), noise, and *SNR*). The effect of the parameters on the µSCM’s performance was carefully investigated, and a novel 4D nomogram was developed. Furthermore, an *SNR* analysis considering noise sources of a low-noise amplifier was performed. By combining the results from the nomogram and the effect of the noise sources from the amplifier circuit, optimum values for the dimensionless parameters were calculated. According to the calculation results, the dominant noise source varied with an increase in the track width ratio to the outer diameter. Seven different samples were fabricated by a single-mask lithography process. The sensitivity of 1612 mV/mT was demonstrated at a 50 Hz input magnetic field, which was better than the previous µSCM (*S_e_* = 6.5 mV/mT) by more than 2 orders of magnitude. Finally, one of the fabricated µSCMs was employed to measure the online power consumption of a personal computer while different types of software were running.

## 1. Introduction

Magnetic field sensors can be realized by different methods such as Hall-effect sensor, magnetodiode, anisotropic magnetoresistance (AMR), giant magnetoresistance (GMR), MEMS Lorentz force, fluxgate, search coil, and superconducting quantum interference device (SQUID) magnetometers [1]. These magnetic field sensors have a wide range of applications. They can measure a wide range of magnetic fields, ranging from certain femto-tesla (using precise SQUID magnetometers) to kilo-tesla (using search coil magnetometers) [2,3]. Depending on the sensor’s type and its requirements, magnetometers can be realized by conventional macro machining (search coil and fluxgate magnetometers), semiconductor process (Hall sensors in silicon and III–V semiconductor), MEMS fabrication processes (MEMS Lorentz force magnetic field sensors), etc. Among these magnetometers, search coil magnetometers (SCMs) have some inherent advantages. An SCM is a passive sensor, where typical power consumption results from the signal conditioning circuit and the amplifier circuit. Therefore, with a proper readout circuit design, they can be considered ultra-low-power sensors, which can operate for several years with small batteries [4]. In addition, SCMs can be used in harsh environments due to their low sensitivity to the ambient temperature compared to other types of semiconductor magnetometers in a wide temperature range [5]. Furthermore, they have the widest dynamic range and good linearity, especially an air-core SCM [3]. Finally, the output voltage of an SCM is proportional both to the input magnetic field amplitude and its frequency. This unique feature of the SCM can be utilized to detect high-frequency magnetic fields.

Zeidi et al. [6] utilized a planar spiral inductor to detect the partial discharge between a sharp needle and a conductive plate. The induced voltage frequency spectrum showed a detectable magnitude in the mega Hz range even 10 cm away from the discharge platform. This indicates that even a very small high-frequency magnetic field can be detected by an SCM due to its unique working principles. Vejella and Chowdhury [7] designed and simulated an ultra-wide-band µSCM with a ferromagnetic core for the GHz frequency range. The induced voltage on the inductor was utilized to change the capacitance on a diaphragm. Although the presented sensor system was not fabricated, a sensitivity of 4.5 aF/0.8 µA/m was achieved according to the reported calculation and FEM simulation results. Despite the lack of fabrication in this work, it is a good option for high-frequency power measurements, such as 5G and Wi-Fi.

Conventional SCMs have a simple fabrication process: wrapping a long thin wire around a magnetic core for several 10,000 turns. The core and wound wire are usually sandwiched between two magnetic field concentrators. Although this simple realization method can achieve a very low noise equivalent magnetic induction (*NEMI*) to several femto-tesla/Hz^0.5^, conventional SCMs are bulky and not CMOS compatible [8,9,10,11,12]. *NEMI* is defined by the ratio of the total noise power spectral density to the sensitivity of an SCM [8]. The sensitivity of an SCM is the ratio of the output voltage to the input magnetic field. Therefore, *NEMI* is inversely proportional to the *SNR* of an SCM [13]. Several works have been reported concerning the optimization and improvement of the performance of SCMs. Grosz and Paperno [14] presented an analytical calculation for the *NEMI* of a conventional SCM. They ignored the stray capacitance of the coil since the frequency of interest was an order of magnitude smaller than the resonance frequency of the inductor. The optimum core and wire diameters were calculated considering the noise sources of the amplifier. The size of their sensor was 60 mm in length and 30 mm in diameter, and they achieved a *NEMI* of 11 pT/Hz^0.5^. Grosz et al. [10] presented a compact, three-axis SCM whose power consumption was 252 µW; therefore, the SCM could operate continuously for 7 years with four 1/2AA lithium batteries. This ultra-low-power feature of the SCM stems from the output signal of the SCM being the induced voltage on the coil without any bias voltage, unlike a Hall-effect sensor. The size of their sensor system was 72 × 69 × 69 mm^3^, and the *NEMI* was 12 pT/Hz^0.5^ at 1 Hz. The power consumption of an SCM can be 2 orders of magnitudes lower than a fluxgate sensor with the same resolution [12]. 

Few works were reported for the miniaturization of conventional SCMs. The previously reported works for µSCM can be classified into fixed and vibrating coils. The vibrating-type µSCMs [15] can overcome the limitation of DC magnetic field detection in terms of the complicated fabrication process and extra circuitry for the actuation of the moveable part. Liu et al., reported a vibrating electromagnetic magnetometer [16]. The magnetometer consisted of a vibrating seesaw plate, which was actuated electrostatically, plus a coil. The applied electrostatic force between electrodes on a glass substrate and the seesaw plate leads to vibration in the vertical direction. When the deposited coil on the moveable seesaw plate is exposed to an external magnetic field, a voltage will be induced on the coil due to Faraday’s induction law. The vibration angle will be maximized at the resonance frequency of the seesaw plate. Therefore, maximum sensitivity can be achieved at the resonance frequency. The maximum sensitivity at the resonance frequency is due to (1) the maximum tilt angle at the resonance frequency (42.404 kHz) and (2) the inner vector product of the normal vector to the coil and the input magnetic field being larger at the resonance frequency. The presented sensor can be used for both AC and DC magnetic field measurement. For the AC magnetic field measurement, electrostatic actuation is not required. Although the fabrication process is not as simple (silicon on glass technology) compared to a Hall sensor, the low power consumption (0.75 µW) of this sensor is a very suitable choice for IoT applications. In addition, sensitivity can be increased at low pressure, which can be achieved by vacuum packaging due to a lower damping coefficient. The vacuum packaging of the vibrating µSCM was reported in reference [17]. The achieved quality factor in this design was 42,000, which is more than one order of magnitude larger than the atmospheric pressure resonator. Liang et al. [18] presented a different actuation mechanism for the vibrating coil compared with reference [17]. They utilized an interdigitated, staggered, comb-drive actuation method instead of a parallel plate type. Since the air damping effect was reduced significantly in this design compared to parallel plate actuation [16], the quality factor of the resonating structure was increased noticeably. The achieved sensitivity and resolution were 468 mV/mT and 6 µT, respectively. The resonance frequency of the structure was 12.65 kHz, and the power consumption was 78 nW, which is much lower than other types of semiconductor magnetometers such as Hall sensors. In reference [19], an out-of-plane resonance induction electromagnetic magnetometer was presented. The in-plane vibration of the S-shaped springs, which are actuated electrostatically by a comb-drive structure, leads to the closed area change of the deposited coils on the springs. Therefore, in the presence of an input DC magnetic field, the output voltage will be induced across the coils due to the enclosed area changes of the coil (this actuation method is not similar to references [15,17], where the angle between the coil and the applied magnetic field varies due to the electrostatic actuation). The sensitive axis is perpendicular to the vibrating structure, while in references [15,17], the external magnetic field must be parallel to the vibrating coil. Vibrating coil structures were also reported in references [20,21] with much higher resonance frequencies (4.361 MHz and 4.33 MHz, respectively) compared with other works. The main disadvantage of the vibrating coil for magnetic field detection is its complicated fabrication process (in some cases, it requires eight-times lithography and four-times DRIE processes [18]) compared with other semiconductor magnetometers that do not have moving parts.

Fixed µSCMs have a simpler fabrication process compared with vibrating coil magnetometers. In addition, they are CMOS compatible and can be realized completely by a CMOS process without any need for post-processing in most cases. In reference [5], two µSCMs with the dimensions of 1 mm × 3 mm were utilized for surface crack detection. The two planar inductors were used to detect the spatial derivative of the magnetic field on the top of the existing crack. The achieved sensitivity with track width = 7 µm, track thickness = 2 µm, and number of turns = 40 was 0.2 mV/mT at 2 kHz. Although the sensitivity of the µSCM was relatively lower than commercial products such as a magnetodiode (MD-130 by Sony) 20 mV/mT, the temperature independency of the µSCM is the inherent advantage of a µSCM over other semiconductor magnetometers. The output signal of the µSCM for crack detection remained unchanged for the temperature range of 30~75 °C, while the output voltage of the magnetodiode dropped by 80% in the mentioned temperature range. Eyre et al. [22] presented a three-axis µSCM. Their work was realized by the post-CMOS process. The vertical coils were connected to the substrate by aluminum hinges, which were permanently deformed to the desired position. The sensor consisted of three spiral inductors, and the sensitivity was calculated by summing up the area of each turn of the coil. Although the thermal noise of DC resistance was calculated, no optimization was performed to achieve the maximum signal-to-noise ratio. The measured and calculated relative sensitivities were 1.3 × 10^−4^ V/T.Hz and 1.6 × 10^−4^ V/T.Hz, respectively. The dimensions of the two-layer stacked inductors were 0.45 mm × 1.5 mm. The measurement was performed in the range of 5 kHz~1.5 MHz, and an almost linear response was reported for this frequency range. Azmi et al. [23] used a planar spiral inductor to perform an FEM simulation to present a µSCM. The size of their sensor was 3.16 mm × 3.16 mm. In the proposed fabrication process, two planar spiral inductors were fabricated separately and bonded by the flip-chip bonding technique. The sensitivity and the noise analyses of the µSCM were not studied in their work. In our previous work [24], a four-layer printed circuit board (PCB) was utilized to realize an SCM with four stacked planar spiral inductors. The mutual inductance [25,26] between the bottom coil and the other three coils was utilized for on-chip calibration purposes, which is very useful for long-term applications. 

The summary of the reported works and their novel ideas are listed in Table 1. To the authors’ best knowledge, few works comprehensively studied the optimization of a µSCM to achieve maximum *SNR* for a low-frequency application such as power monitoring of home appliances in smart buildings. In this work, *SNR* optimization was achieved by introducing three dimensionless parameters. An analytical method was presented to calculate the *SNR*. The presented method included all layout parameters of the planar spiral inductor. To calculate the noise of the µSCM, the thermal noise of the coil and all noise sources of the utilized instrumentation amplifier were included in the *SNR* calculation. 

## 2. Materials and Methods

The working principle of the µSCM is based on Faraday’s law of induction [3]. When a fixed inductor is exposed to an alternating magnetic field, an AC voltage is induced on the coil. The induced voltage is proportional to the derivative of the input magnetic field, as shown in Figure 1a. The output voltage of the µSCM can be calculated by Equation (1):(1)$$V_{o}=A_{e}\frac{dB_{i}}{dt}$$
where *V_o_*, *B_i_*, and *A_e_* are the inducted output voltage, input magnetic field, and the effective area of the inductor, respectively. If the input magnetic field *B_i_* is equal to $B_{m}\sin\left(2\pi ft\right)$, then the output voltage for a fixed coil can be calculated by Equation (2):(2)$$V_{o}=A_{e}B_{m}2\pi f\cos\left(2\pi ft\right)$$
where *f*, *B_m_*, and *t* are the frequency of the input magnetic field, the amplitude of the input magnetic field, and time, respectively. To miniaturize the SCM and realize it through the microfabrication process, a square planar spiral inductor was utilized in this work. This inductor can be defined by its layout parameters: track width (*W*), track spacing (*S*), track thickness (*h*), outer diameter (*D_o_*), and inner diameter (*D_i_*). The planar spiral inductor can be approximated by closed-loop areas [25,26], as shown in Figure 1b. The flowchart of the optimum design methodology is shown in Figure 1c. The three dimensionless numbers were defined to achieve more general results. The effect of these parameters was summarized in a 4D nomogram. By combining the results from the nomogram and the effect of the noise sources from the amplifier circuit, optimum values for the mentioned dimensionless parameters could be calculated. 

The number of turns (*N*) of the planar inductor is a function of the aforesaid geometrical layout parameters and can be formulated by Equation (3):(3)$$N=\frac{D_{o}-D_{i}+2S}{2\left(W+S\right)}\;,\;N\geq 1\;\;\&\;\;D_{o}\geq D_{i}+2W$$

In this work, a square planar inductor was used to enable the maximum enclosed area in a fixed die size. The effective area of the square planar inductor is calculated by Equation (4):(4)$$A_{e}=\overset{N}{\underset{i=1}{\sum }}A_{i}$$
where *A_i_* is the area of *i*th turn, which can be formulated by Equation (5) for the square-shape spiral planar inductor.
(5)$$A_{i}={\left[D_{i}+2W+2\left(W+S\right)\left(i-1\right)\right]}^{2}$$

Therefore, the effective area of the square planar inductor can be calculated by Equation (6):(6)$$\begin{array}{c} A_{e}=\frac{4{\left(W+S\right)}^{2}}{3}N^{3}+2\left(W+S\right)\left(D_{i}+W-S\right)N^{2}\\ \;+\left[\frac{2{\left(W+S\right)}^{2}}{3}+\left(D_{i}+2W\right)\left(D_{i}-2S\right)\right]N \end{array}$$

By substituting Equation (3) into Equation (6), the effective area can be calculated as a function of layout parameters by Equation (7):(7)$$\begin{array}{c} A_{e}=\frac{{\left(D_{o}-D_{i}+2S\right)}^{3}}{6\left(W+S\right)}+\frac{\left(D_{i}+W-S\right){\left(D_{o}-D_{i}+2S\right)}^{2}}{2\left(W+S\right)}\\ \;+\frac{\left(W+S\right)\left(D_{o}-D_{i}+2S\right)}{3}\\ \;+\frac{\left(D_{i}+2S\right)\left(D_{i}-2S\right)\left(D_{o}-D_{i}+2S\right)}{2\left(W+S\right)} \end{array}$$

The sensitivity (*S_e_*) of the µSCM can be defined by dividing the maximum induced voltage by the maximum input magnetic field [22], which is formulated by Equation (8): (8)$$S_{e}=\frac{V_{m}}{B_{m}}=2\pi fA_{e}\;\frac{V}{T}$$
where *V_m_* is the maximum induced voltage at the maximum input magnetic field. According to Equations (7) and (8), the sensitivity monotonically increases by increasing *D_o_* and decreasing *w* and *s*. Increasing *D_o_* will lead to an increase in the size of the µSCM, which is usually limited in microfabrication. However, in a fixed die size, decreasing the track width and spacing will increase the DC resistance of the coil. The DC resistance of the µSCM can generate thermal noise, of which the corresponding spectral density (*e_nR_*) can be determined as follows, i.e., Equation (9):(9)$$e_{nR}=\sqrt{4k_{B}TR}\;\frac{V}{\sqrt{\mathrm{Hz}}}$$
where *k_B_*, *T*, and *R* are the Boltzmann constant, the temperature in Kelvin, and resistance, respectively. 

The DC resistance of the inductor can be calculated by Equation (10) (notably, the skin effect is ignored in Equation (10) since the focus of this work was on low-frequency magnetic fields, such as the 50 Hz magnetic field around home appliances):(10)$$R=\rho \frac{L}{Wh}$$
where *ρ* and *L* are the inductor material’s resistivity and total length of the inductor, respectively. The total length of the spiral coil can be approximated by Equation (11):(11)$$L=\overset{N}{\underset{i=1}{\sum }}L_{i}$$
where *L_i_* is the length of the *i*th turn. For a square spiral inductor, *L_i_* can be calculated by Equation (12):(12)$$L_{i}=4\left[D_{i}+2W+2\left(W+S\right)\left(i-1\right)\right]$$

Therefore, the total coil’s length can be formulated by Equation (13):(13)$$L=4N\left(D_{i}-2S\right)+4\left(W+S\right)\left(N^{2}+N\right)$$

By substituting Equations (3) and (13) into Equation (10), the resistance of the square planar inductor can be expressed by Equation (14):(14)$$R=\frac{\rho \left(D_{o}+D_{i}+2W\right)\left(D_{o}-D_{i}+2S\right)}{Wh\left(W+S\right)}$$

The *SNR* of the µSCM is the ratio of the output voltage to the noise voltage, which is expressed by Equation (15):(15)$$SNR=\frac{output\;voltage}{noise\;voltage}=\frac{V_{o\_rms}}{e_{nR}\times \sqrt{BW}}=\frac{2\pi fA_{e}B_{m}}{\sqrt{4K_{B}TR}\times \sqrt{BW}}$$
where *V_o_rms_* and *BW* are the root mean square (rms) value of the output voltage and the noise measurement bandwidth, respectively. The resolution (minimum detectable magnetic field, commonly called *NEMI*) is the ratio of noise to sensitivity [14,22,27]. Here, we assume normalized *SNR* (*SNR_n_*) for a µSCM, which is the inverse of *NEMI* and can be defined by Equation (16):(16)$$SNR_{n}=\frac{1}{NEMI}=\frac{sensitivity}{noise}=\frac{S_{e}}{e_{nR}}=\frac{2\pi fA_{e}}{\sqrt{4K_{B}TR}}\;\frac{\sqrt{\mathrm{Hz}}}{T}$$

Therefore, *SNR_n_* is a function of geometrical layout parameters, coil material resistivity, and temperature. *SNR_n_* can be expressed by Equation (17):(17)$$SNR_{n}=SNR_{n}\left(K_{B},\;T,\;BW,\;\rho ,\;h,\;D_{o},\;D_{i},\;W,\;S\right)$$

The resistivity and thickness of the metal layers are usually predetermined in the fabrication process (e.g., a CMOS foundry process). Therefore, *SNR_n_* can be rewritten by Equation (18):(18)$$SNR_{n}=\pi \sqrt{\frac{h}{K_{B}T\rho }}SNR^{*}\left(D_{o},\;D_{i},W,\;S\right)$$
where *SNR** is defined by Equation (19). Thus, the goal of this work was to find optimal values for geometrical layout parameters to achieve maximum *SNR**.
(19)$$SNR^{*}=\frac{A_{e}\sqrt{W\left(W+S\right)}}{\sqrt{\left(D_{o}+D_{i}+2W\right)\left(D_{o}-D_{i}+2S\right)}}$$

According to the Buckingham π theorem, dimensional analysis can be used to identify critical normalized parameters [28,29] and the three dimensionless parameters are defined to have more generalized results. These parameters are defined in Equation (20):(20)$$d_{i}=\frac{D_{i}}{D_{o}},\;w=\frac{W}{D_{o}},\;s=\frac{S}{D_{o}}$$

As mentioned in Equation (3), $D_{o}\geq D_{i}+2W$; so, $d_{i}+2w\leq 1$. By substituting Equation (20) into Equation (19), *SNR** can be rewritten by Equation (21):(21)$$\frac{SNR^{*}}{D_{o}{}^{2}}=\frac{A_{e}\sqrt{w\left(w+s\right)}}{\sqrt{\left(1+d_{i}+2w\right)\left(1-d_{i}+2s\right)}}$$

By substituting Equations (7) and (20) into Equation (21) and dividing *SNR** by the die area, which is *D_o_*^2^, Equation (22) can be derived. Equation (22) can be used to study the effect of the three dimensionless parameters, which can provide more general results compared to pure geometrical layout parameters.
(22)$$\begin{array}{c} \frac{SNR^{*}}{D_{o}{}^{2}}=\frac{\sqrt{w\left(w+s\right)}}{\sqrt{\left(1+d_{i}+2w\right)\left(1-d_{i}+2s\right)}}[\frac{{\left(1-d_{i}+2s\right)}^{3}}{6\left(w+s\right)}\\ \;+\frac{\left(d_{i}+w-s\right){\left(1-d_{i}+2s\right)}^{2}}{2\left(w+s\right)}+\frac{\left(w+s\right)\left(1-d_{i}+2s\right)}{3}\\ \;+\frac{\left(d_{i}+2w\right)\left(d_{i}-2s\right)\left(1-d_{i}+2s\right)}{2\left(w+s\right)}] \end{array}$$

Here, we try to obtain the optimized maximum *SNR**/*D_o_*^2^ for an improved *SNR*. The effect of the three dimensionless parameters is studied in the following sections.

### 2.1. Effect of d_i_, While w and s Are Fixed

The effect of *d_i_* for different fixed *w* = *s* is shown in Figure 2a. Without a loss of generality, this figure is plotted for *D_o_* = 1 mm and *D_o_*^2^ = 1 mm^2^, considered to be unity for simplicity. As shown in Figure 2a, two regions were determined by increasing *d_i_*. In region 1, *SNR**/*D_o_*^2^ did not change substantially. This is because, by increasing *d_i_*, which decreased the number of turns, both thermal noise and sensitivity decreased. Although sensitivity and noise decreased simultaneously, the ratio of sensitivity to noise was almost constant in region 1. Therefore, there was no significant change in the *SNR**/*D_o_*^2^ value. In region 2 of Figure 2a, sensitivity reduction was dominant compared to noise reduction. Thus, *SNR**/*D_o_*^2^ started to drop sharply. An important result can be concluded from Figure 2a: almost the same *SNR**/*D_o_*^2^ can be achieved with a reduced number of turns, as shown in Figure 2b. This result was beneficial in reducing the DC resistance of the µSCM. Furthermore, reducing the number of turns led to a reduction in the coupling capacitance of the spiral planar inductor [30]. Therefore, the resonance frequency and the bandwidth of the µSCM increased without a noticeable change in the *SNR**/*D_o_*^2^.

### 2.2. Effect of w = s, While d_i_ Is Fixed

If the normalized track width (*w*) and the normalized spacing (*s*) change simultaneously with a fixed normalized inner diameter (*d_i_*), there is a slight change in the value of *SNR**/*D_o_*^2^. This issue is illustrated in Figure 3, where *D_o_* = 1 mm and *D_o_*^2^ = 1 mm^2^ is considered to be unity for simplicity. The intuitive explanation for this issue is that the ratio of sensitivity and noise of the µSCM is almost constant when *w* and *s* change simultaneously.

### 2.3. Effect of w, While d_i_ and s Are Fixed

The variation of *w*, while *d_i_* and *s* are fixed, has a substantial effect on *SNR**/*D_o_*^2^, as shown in Figure 4. Similar to the previous two figures, *D_o_* = 1 mm and *D_o_*^2^ is considered to be unity for simplicity without a loss of generality. Some results can be concluded from Figure 4:-Smaller *s* leads to greater *SNR**/*D_o_*^2^. In other words, the smaller track spacing is preferred to achieve higher *SNR* for a µSCM. This statement is valid for the low-frequency range (at least one order of magnitude smaller than the resonance frequency of the inductor [14]), where the coupling capacitance of the coil can be ignored.-By increasing *w* in region 1, noise reduction has a dominant effect on *SNR**/*D_o_*^2^ compared with sensitivity reduction. Therefore, greater *SNR**/*D_o_*^2^ can be achieved in this region by increasing *w*. It is interesting to note that region 1 is the practical region because the output voltage in this region can be detected or amplified by available commercial readout circuits.-By increasing *w* in region 2, noise and sensitivity reduction are approximately the same. Therefore, there is no significant change in the value of *SNR**/*D_o_*^2^. Thus, *SNR**/*D_o_*^2^ tends to saturate in region 2. Although *SNR**/*D_o_*^2^ offers a higher value in region 2, this region is not an efficient region for practical applications. The induced voltage in region 2 can be smaller than the input-referred noise of an amplifier in most cases. Therefore, it is advisable to choose a proper value for *w* in region 1. This is further explained in Section 2.5. 

### 2.4. Novel 4D Nomogram for a General Guide Design of the µSCM

By combining the results of the previous sections, we proposed a novel 4D nomogram that describes *SNR**/*D_o_*^2^ as a function of the normalized inner diameter (*d_i_*), normalized track width (*w*), and normalized spacing (*s*), as shown in Figure 5a. In this nomogram, *D_o_* = 1 mm and *D_o_*^2^ = 1 mm^2^ is considered to be unity for simplicity without a loss of generality. This 4D nomogram can be used as a general guideline for designers to design an optimized µSCM with a square planar inductor. After choosing the maximum available die size, the proper inner diameter can be chosen from this figure. This issue is illustrated more clearly in Figure 5b, which is the 2D perspective of Figure 5a. As shown in Figure 5b, *SNR**/*D_o_*^2^ is almost constant for a specific range of *d_i_* (almost *d_i_* < 0.5). As described previously, *s* must be set to the minimum value according to the minimum track spacing in the fabrication process. Finally, *w* should be set to the maximum value while considering the noise of the designed µSCM is greater than the input-referred noise of the readout circuit. Although the theoretically higher value of *SNR**/*D_o_*^2^ can be achieved by increasing *w*_,_ as shown in Figure 5a, in practical terms, the output-induced voltage at a large value of *w* will be too small to be measured or amplified by available commercial electrical components or instruments. Therefore, the noise of the readout circuit must be considered when choosing a proper value for *w*. Therefore, taking this into account, the noise sources of the utilized amplifier are complementary to the 4D nomogram, as shown in Figure 1c. Further information about this issue can be found in Section 2.5. 

The exponents of important parameters were extracted and are listed in Table 2. It is interesting to note that noise (*e_nR_*), sensitivity (*S_e_*), and *SNR_n_* increased by enlarging *D_o_*, but with different exponents, as seen in Table 2. Notably, fabrication of the µSCM using a metal with higher track thickness and lower resistivity can further improve the *SNR_n_* (*SNR_n_* ∝ $h^{0.5}\times \rho^{-0.5}$). Therefore, it is better to use the maximum possible metal thickness and lowest resistivity in the fabrication of the µSCM (in the low-frequency range where the skin effect can be ignored). In this work, aluminum with a 2 µm thickness was utilized for the fabrication of the µSCM due to the practical limitations in the fabrication process. Furthermore, a metal with greater thickness and lower resistivity such as gold or copper can further improve the *SNR_n_*.

### 2.5. Effect of Amplifier Noise Parameters on SNR

The noise model of the µSCM integrated with an instrumentation amplifier is illustrated in Figure 6a. The noise sources of the instrumentation amplifiers should also be considered in the calculation of *SNR_n_*. The total noise spectral density can be calculated by Equation (23) [31]:(23)$$e_{nT}=\sqrt{2{\left(i_{n-in}\frac{R}{2}\right)}^{2}+e_{nR}^{2}+e_{n-in}{}^{2}+{\left(\frac{e_{n-out}}{G}\right)}^{2}}\frac{V}{\sqrt{\mathrm{Hz}}}$$
where *e_nT_*, *i_n-in_*, *e_n-in_*, *e_n-out_*, and *G* are total input-referred noise spectral density, input current noise, input voltage noise, output noise, and the gain of the instrumentation amplifier, respectively. As shown in Figure 6a, the noise sources can be classified into sensor noise (*e_nR_*) and amplifier noise (*e_n-amp_*). It is notable that the higher resistance value of the µSCM not only increased the thermal noise of the sensor, but it also increased the amplifier noise because of its input current noise. This statement can be verified by Equation (23). 

The effect of the normalized track width (*w*) on *SNR**/*D_o_*^2^ without considering the amplifier noise was presented in previous sections. It was shown that increasing *w* leads to a higher *SNR*, without considering the noise of the amplifier circuit. The comparison between the noise of the µSCM and the utilized instrumentation amplifier is illustrated in Figure 6b. As illustrated in this figure, at small values of *w*, the amplifier noise (*e_n-amp_*) is greater than the µSCM noise (*e_nR_*). In this region, the current noise of the amplifier is the dominant noise source. By increasing *w*, current noise becomes less significant and the sensor noise becomes the dominant noise source. By further increasing *w*, current noise and sensor noise gradually become negligible compared to the voltage noise of the amplifier. In this region, the voltage noise is the dominant noise source, and increasing the *w* does not help to enhance the *SNR*. Therefore, there is a critical *w* that leads to maximum *SNR*. This issue is proven analytically and experimentally in Section 3.2. 

## 3. Results and Discussion

### 3.1. Fabrication

To validate the presented analyses experimentally, different µSCMs were fabricated by low-cost, single-mask fabrication processes on a 4-inch silicon wafer at clean rooms of Nanosystem Fabrication Facility of Hong Kong University of Science and Technology. The fabrication process is shown in Figure 7. The fabrication started with cleaning the silicon wafer with piranha solution (H_2_SO_4_:H_2_O_2_, 10:1), followed by thermal oxidation and 2 µm aluminum deposition with sputtering. Then, HPR504 positive photoresist was coated and patterned to act as a mask layer for the aluminum layer. Afterward, an Oxford aluminum dry etcher (Oxford Instruments, Oxfordshire, UK) was employed to etch the aluminum layer, followed by deionized water rinsing to eliminate the chlorine corrosion. After drying, oxygen plasma was utilized to strip the photoresist. Finally, the fabricated µSCM was mounted on a PCB and wire bonded to the copper layer on the PCB. Seven different samples with different layout parameters were fabricated and characterized. The SEM picture of one of the fabricated samples with an outer diameter of 1 mm is illustrated in Figure 7f.

### 3.2. SNR Measurement

In our previous work [32], we studied the effect of different amplifiers’ noise parameters on the *SNR* of a µSCM. In this work, a zero-drift, precise instrumentation amplifier (INA188, Texas Instruments, Dallas, TX, USA) was utilized to amplify the induced voltage on the µSCM. In addition to the excellent noise feature of this amplifier, it had a very high input impedance (100 GΩ|| 6 pF for differential mode and 100 GΩ|| 9.5 pF for common mode) that made it a suitable choice for the µSCMs in this work (*R* < 30 kΩ). The schematic of the amplifier circuit is shown in Figure 8a. To characterize fabricated µSCMs, they were placed inside a uniform magnetic field generated by a commercial Helmholtz electromagnet (WD-50, YP Magnetic Technology Co. Ltd., Changchun, China), as shown in Figure 8a. Noise measurement was performed by a spectrum analyzer (HP DSA Dynamic Signal Analyzer 35665A, Keysight Technologies, Santa Rosa, CA, USA) at a zero input magnetic field. To experimentally study the effect of the outer diameter on the *SNR* and compare it with the presented theoretical model, five different samples (with *D_o_* = 1 mm~5 mm, *D_i_* = 200 µm, *S* = 1 µm, *W* = 3 µm, *h* = 2 µm) were fabricated and characterized.

The sensitivity of the samples was measured at a 50 Hz magnetic field, and the noise measurement was performed up to 800 Hz by a spectrum analyzer. The sensitivity and the noise of the five mentioned µSCMs were measured by the aforesaid instruments. The measurement and calculation results are shown in Figure 8b. Notably, rms voltage was utilized for noise and output voltage measurements. As shown in this figure, the presented calculation method can effectively predict the measured *SNR*. Furthermore, the *NEMI*, which is related to the minimum detectable field intensity, of the fabricated samples was measured from the noise and sensitivity measurement results. The *NEMI* of 1020 nT/Hz^0.5^~16.2 nT/Hz^0.5^ was achieved for *D_o_* = 1 mm~5 mm at 50 Hz. To study the effect of *w* on the maximum achievable *SNR*, three different samples, with fixed *D_o_* = 1 mm; *S* = 1 µm; *D_i_* = 200 µm; *h* = 2 µm; and *W* = 1 µm, 3 µm, and 5 µm, were fabricated and characterized. Notably, the metal width was not perfectly uniform in the fabrication process. According to our measurement results at different points, the maximum line width error was 118 nm, as shown in Figure 7f. This error caused a maximum of 7.1% discrepancy between the µSCM’s resistance calculation and measurement. This error had little effect on the optimum design of the µSCM since the *SNR_n_* slope around the optimum dimensions was small. The calculation and measurement results for these three samples are shown in Figure 8c. Notably, all of the *SNR* values were normalized to the maximum calculated *SNR* value in this figure. As shown in this figure, *SNR* started to decrease when *w* increased. This is because the noise reduction of the µSCM did not play a pivotal role in *SNR* enhancement, and *SNR* decreased because of sensitivity reduction by increasing *w*. Notably, the number of turns decreased by increasing *w*, which led to an effective area reduction of the µSCM. To further study the effect of *w* on *SNR*, a new dimensionless parameter was defined as the ratio of the amplifier noise over sensor noise by Equation (24):(24)$$NR=\frac{e_{n-amp}}{e_{nR}}=\frac{\sqrt{2{\left(i_{n-in}\frac{R}{2}\right)}^{2}+e_{n-in}{}^{2}+{\left(\frac{e_{n-out}}{G}\right)}^{2}}}{\sqrt{4K_{B}TR}}$$
where *NR* and *e_n-amp_* are the noise ratio of the amplifier over the sensor noise and total amplifier input-referred noise, respectively.

The effect of *w* on *NR* is shown in Figure 8c. At small values of *w*, both sensor and amplifier noise were fairly large, but amplifier noise was larger due to its current noise. Although sensitivity is greater at a small *w*, the large noise from the instrumentation amplifier and µSCM resulted in a lower *SNR*. It is interesting to note that the noise of the amplifier equaled the sensor noise twice, as shown in Figure 8c. At the first intersection, which occurred at a smaller *w*, the total noise from the amplifier and sensor was dominant and *SNR* could still be improved by increasing *w* despite a decrease in the sensitivity. However, after the second intersection of two noises, the noise of the µSCM started to drop sharply and sensitivity reduction became the dominant factor in determining *SNR*. After the second intersection, the noise of the amplifier became the dominant noise source and the sensor noise reduction was no longer beneficial for *SNR* enhancement. Therefore, the critical *w* was determined at the second intersection of two noises. 

A comparison between one of the fabricated samples and several commercial products is listed in Table 3. The sensitivity of the fabricated µSCM (1612 mV/mT) was better than the previous µSCM (*S_e_* = 6.5 mV/mT) [22] by 2 orders of magnitude at a 50 Hz magnetic field. In addition, the performance of the fabricated µSCM was better than many commercial Hall-effect sensors for energy monitoring of home appliances. The *SNR* was measured for the fabricated µSCMs and calculated for the Hall-effect sensors (according to their datasheets) at 1 mT, 50 Hz magnetic field, and 800 Hz noise bandwidth. Although the size of the presented µSCMs was larger than a Hall-effect sensor, it did not need offset cancelation, bias voltage, or temperature compensation [5] (especially for industrial applications to monitor power consumption with a wide range of operating temperatures). Furthermore, the sensitivity of a µSCM is proportional to the input magnetic field’s frequency.

### 3.3. Power Measurement

The characterization of the fabricated samples was performed at a 50 Hz magnetic field. Power measurement of home appliances can be one of the applications of the µSCMs. One of the fabricated samples (*D_o_* = 5 mm, *W* = 3 µm, *S* = 1 µm, *h* = 2µm, *D_i_*= 200 µm) was employed to measure the power consumption of a personal computer (OptiPlex 7050, Dell Technologies, Round Rock, TX, USA), while different kinds of software were running on the PC. The generated magnetic field around the phase line was proportional to the current passing through the line. Therefore, by measuring the magnetic field around the phase line, the current could be calculated. After measuring the current of the phase line, the power consumption of the PC could be calculated by multiplying the measured current by the line voltage (220 V for our experiment). Figure 9a shows the block diagram of the PC’s power measurement setup. Figure 9b illustrates the flow diagram for the power measurement setup with its required components. The fabricated µSCM and the phase line (which was connected to the PC) were placed between two commercial ferrites. Notably, the commercial ferrites were used as a magnetic field concentrator (MFC). The µSCM was connected to the readout circuit; the output of the readout circuit was connected to the ADC pin of an open-source wireless MCU (Arduino Yun). The digitized data were transmitted wirelessly to a laptop through a Wi-Fi router, and then the data were stored on the laptop, as shown in Figure 9b. Notably, a precise commercial multimeter meter (Agilent 34401A, Keysight Technologies, Santa Rosa, CA, USA) was utilized to calibrate the digitized data. The measured power of a PC is illustrated in Figure 9c. The consumed power of the PC varies when different kinds of software perform a simulation. This issue is illustrated in Figure 9c. As shown in this figure, the power consumption of the PC changes when heavy software such as MATLAB (version R2019a, from MathWorks, Natick, MA, USA) or Ansys Maxwell (version 16.0.2, from Ansys Inc., Canonsburg, PA, USA) performs an operation or FEM simulation on the PC. 

## 4. Conclusions

We conducted dimensional and scaling analyses for a µSCM and proposed a novel 4D nomogram for the design optimization of a µSCM’s *SNR* for low-frequency applications, such as power measurement of home appliances. The three dimensionless parameters (track width, spacing, and inner diameter divided by the outer diameter) were defined and utilized to perform the theoretical analysis for the sensor’s key performance indices. The effect of these parameters was studied in detail. It was concluded that the sensitivity, the noise, and the *SNR* of a µSCM without an amplifier circuit are proportional to *D_o_*^3^, *D_o_*^1^, and *D_o_*^2^, respectively. Furthermore, the noise sources of an instrumentation amplifier were included in the analysis. It was proven that either noise or sensitivity could be the dominant factor in determining the *SNR* of the sensor for a different range of dimensionless parameters.

Moreover, the dominant noise source varied from sensor to amplifier by changing *w* (the ratio between track width to outer diameter). The critical values for dimensionless parameters to achieve maximum *SNR* were calculated. It was shown that the noise of the amplifier equaled that of the sensor at two values of *w*. The second intersection of the abovementioned noises was the critical value for *w* to achieve maximum *SNR*. Seven different samples were fabricated by a single-mask lithography process. The noise (with an 800 Hz measurement bandwidth) and sensitivity of the fabricated samples were measured at zero input and a 50 Hz magnetic field, respectively. The measurement results for sensitivity, noise, and *SNR* (at 1 mT a 50 Hz magnetic field) were compared with the theoretical analysis. The discrepancy between *SNR* calculation and measurement was 12.8% and 1.5% for *D_o_* = 1 mm and *D_o_* = 5 mm, respectively. It was observed that *SNR* increased by increasing the outer diameter of the µSCM. The sensitivity of 1612 mV/mT and *SNR* (at 1 mT and noise bandwidth of 800 Hz) of 63.5 dB with *D_o_* = 5 mm was better than the previously reported µSCM (*S* = 6.5 mV/mT) by more than 2 orders of magnitude [22] at a 50 Hz magnetic field. In addition, one of the fabricated µSCMs with excellent sensitivity was used to measure the power consumption of a PC, while different computer programs performed an operation or FEM simulation. In summary, µSCM will be useful for real-time monitoring of smart buildings to achieve a significant reduction of carbon emission and much high-energy efficiency in the era of the Internet of Things. 

## Figures and Tables

**Figure 1 micromachines-13-01342-f001:**
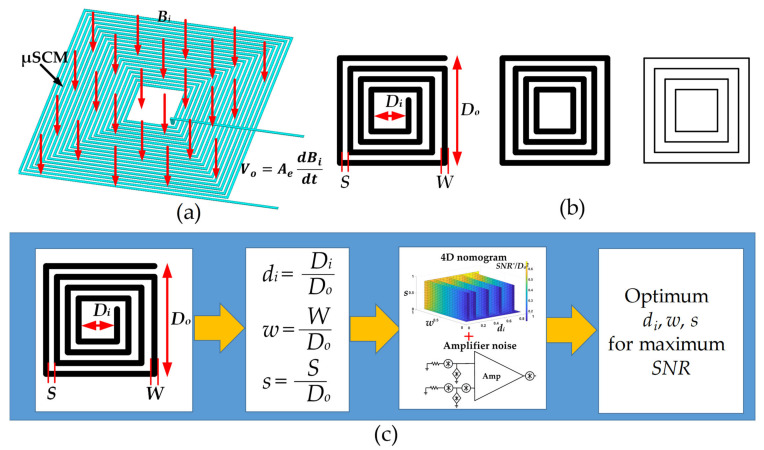
(**a**) Working principle of µSCM. (**b**) Approximation of a spiral inductor with closed areas and its layout parameters. (**c**) Optimum design methodology to achieve maximum *SNR*.

**Figure 2 micromachines-13-01342-f002:**
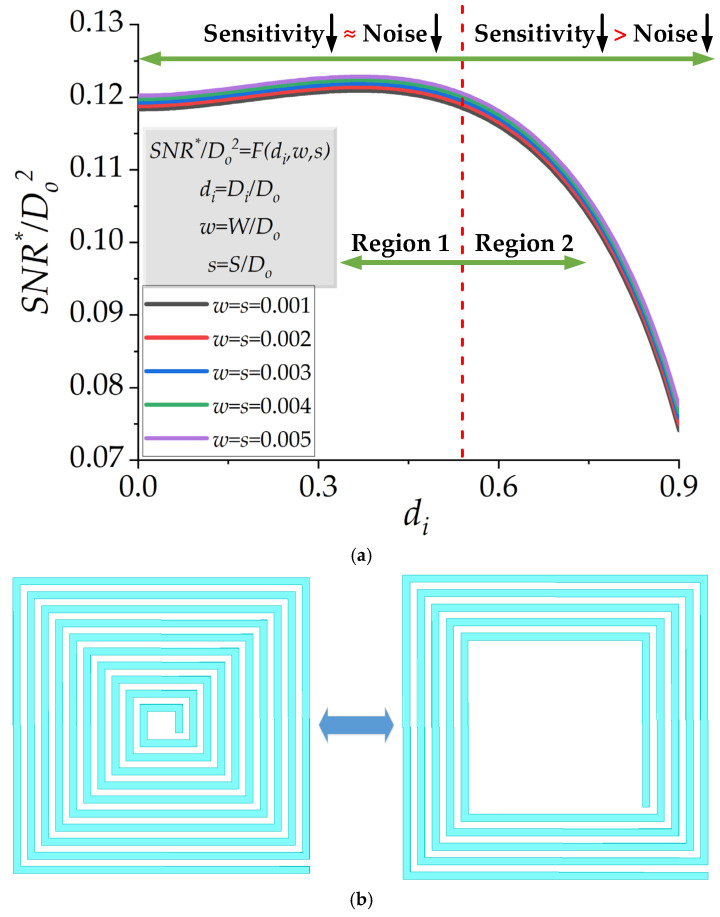
(**a**) *SNR**/*D_o_*^2^ as a function of normalized *D_i_* (*d_i_*) shows the existence of the critical size of the normalized inner diameter *d_i_* (~0.5). (**b**) Dimensional analysis proves that the same *SNR**/*D_o_*^2^ can be achieved with a reduced number of turns, which can improve the quality factor and bandwidth of the µSCM.

**Figure 3 micromachines-13-01342-f003:**
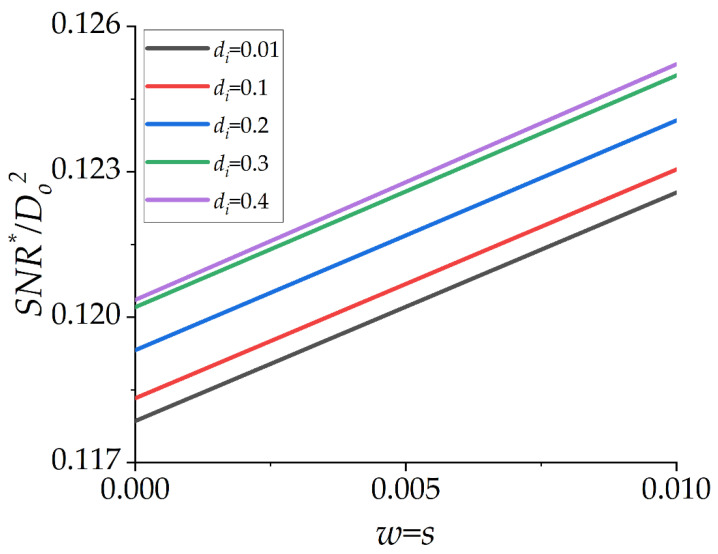
*SNR**/*D_o_*^2^ as a function of the normalized track width *w* (=*s*) for different normalized inner diameters (*d_i_*). The value of *SNR**/*D_o_*^2^ changes slightly, while *w* and *s* are identical and change simultaneously.

**Figure 4 micromachines-13-01342-f004:**
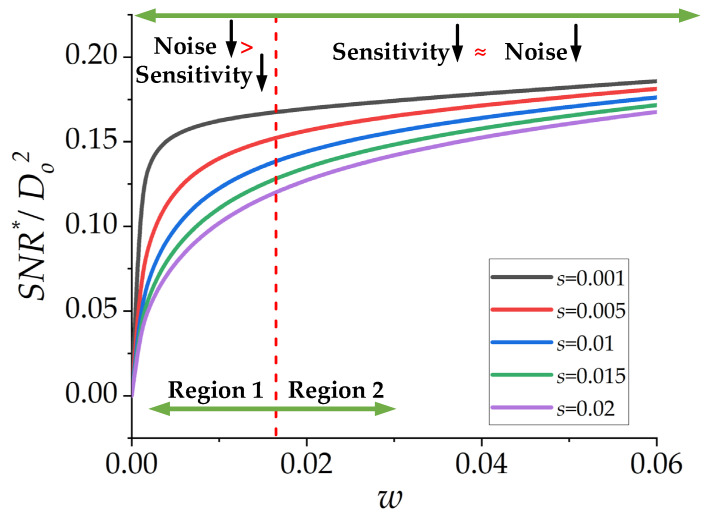
*SNR**/*D_o_*^2^ as a function of the normalized track width (*w*) for different *s* while *d_i_* = 0.01.

**Figure 5 micromachines-13-01342-f005:**
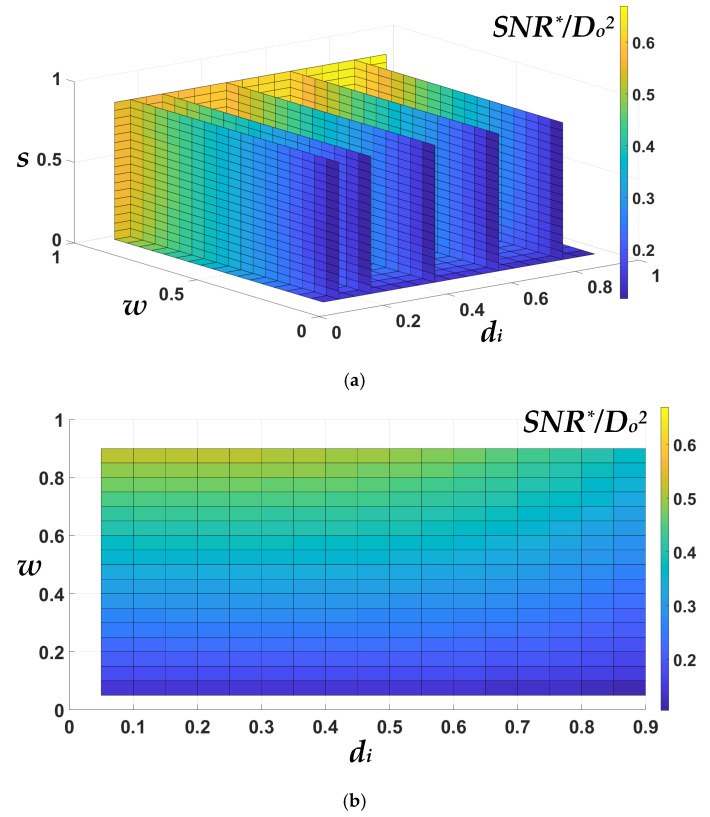
(**a**) The 4D nomogram for *SNR**/*D_o_*^2^ as a function of *d_i_*, *w*, and *s* provides a novel design guideline for µSCM design optimization. (**b**) The 2D view of part (**a**) shows the existence of a critical normalized inner diameter (*d_i_*_,c_).

**Figure 6 micromachines-13-01342-f006:**
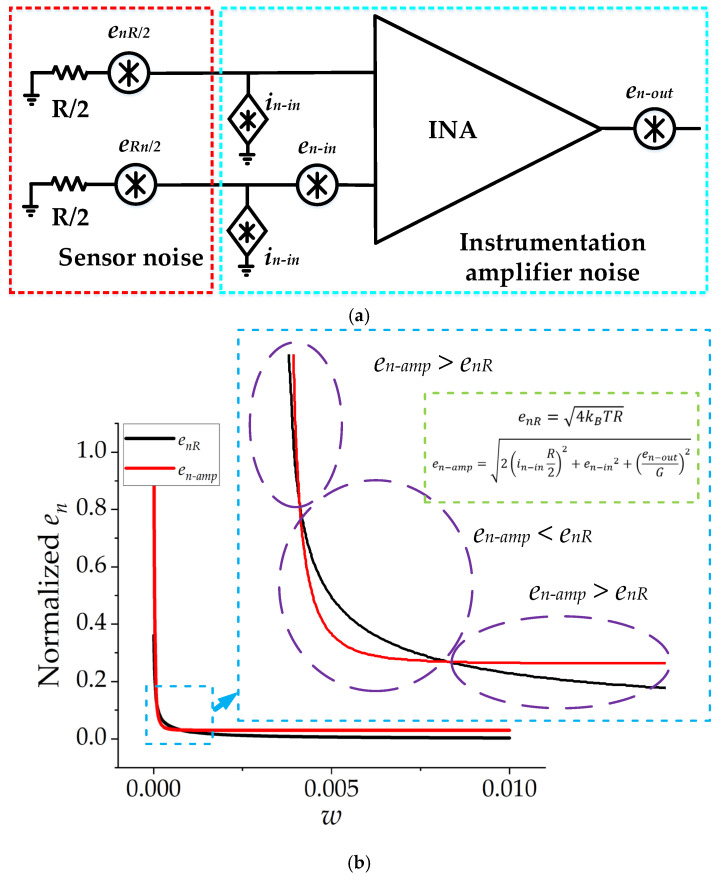
(**a**) The noise model of the µSCM integrated with an instrumentation amplifier at a low-frequency range. (**b**) The normalized noise power spectral density of sensor noise and instrumentation amplifier (INA188) noise as a function of *w* (*D_o_* = 1 mm, *d_i_* = 0.2, *s* = 0.001).

**Figure 7 micromachines-13-01342-f007:**
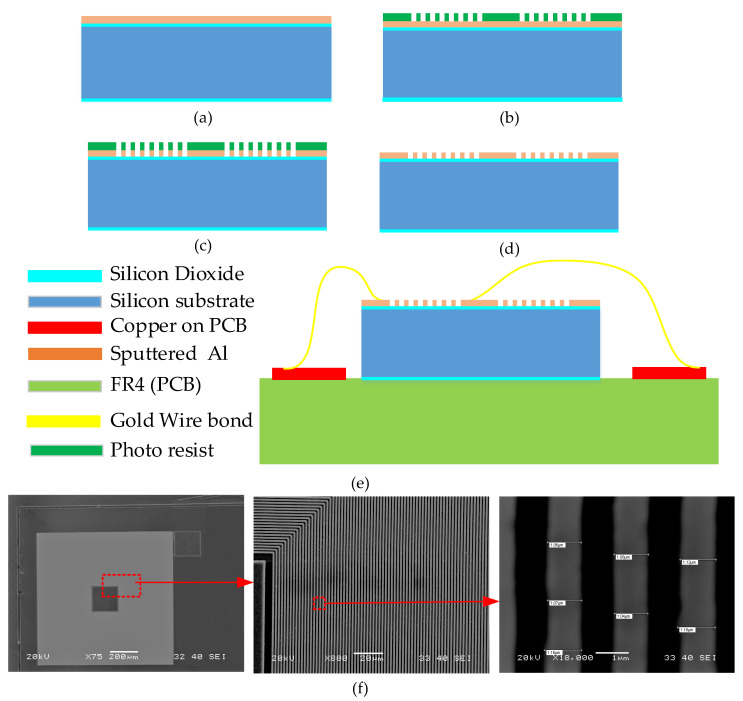
Fabrication processes of single-layer µSCM using a silicon wafer. (**a**) Thermal oxidation and Al sputtering. (**b**) Photoresist patterning by contact photolithography. (**c**) Al dry etch. (**d**) Photoresist removal by dry method. (**e**) Wafer dicing and wire bonding. (**f**) SEM pictures of one of the fabricated µSCMs.

**Figure 8 micromachines-13-01342-f008:**
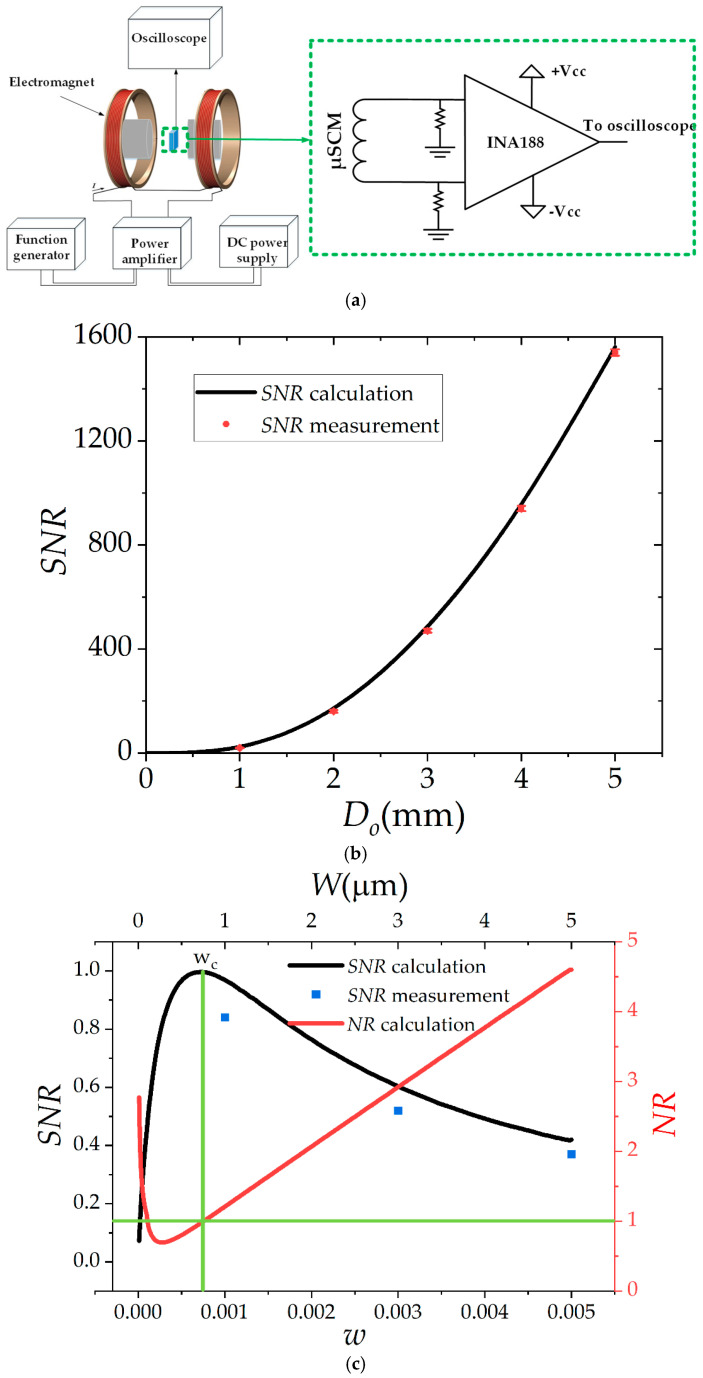
(**a**) The schematic of the measurement setup and the amplifier circuit. (**b**) Comparison between calculation and measurement results for *SNR*. (**c**) Normalized *SNR* and noise ratio (*NR* = *e_n-amp_*/*e_nR_*) as a function of *w* with the noise parameters of INA188 for three different fabricated samples. All *SNR* values were normalized to the maximum calculated *SNR*.

**Figure 9 micromachines-13-01342-f009:**
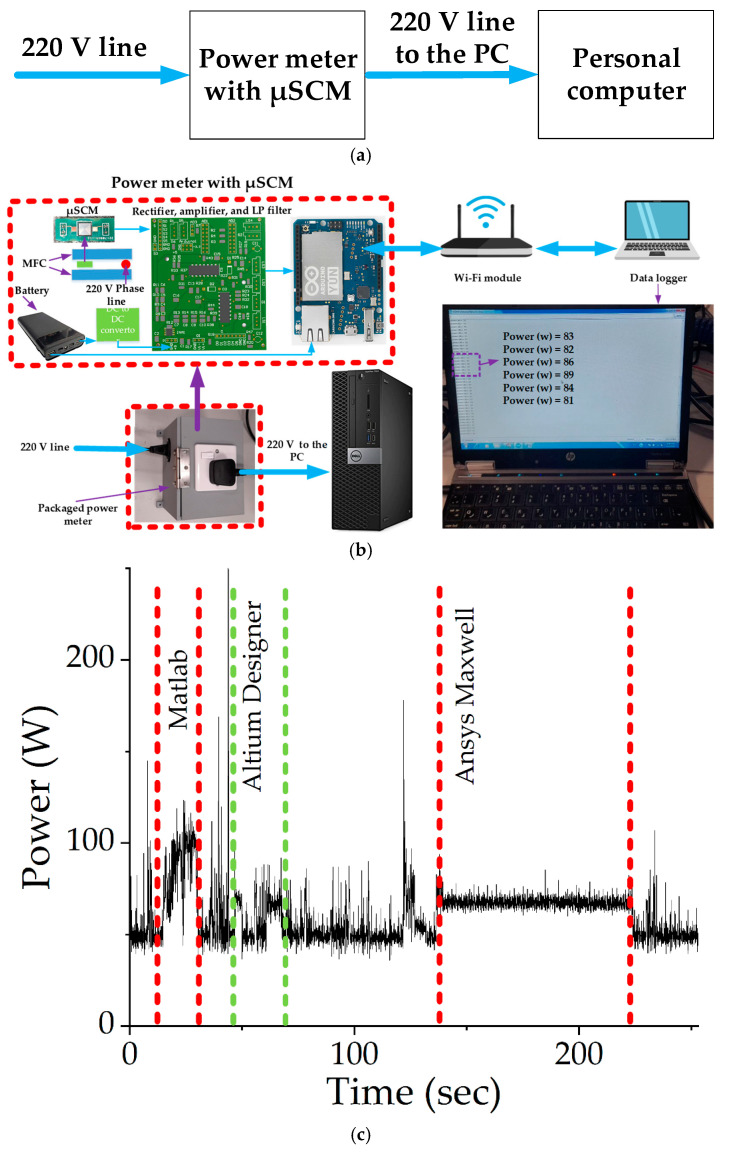
(**a**) The block diagram of the power measurement setup. (**b**) A personal computer power measurement setup with its required components. (**c**) The power of a PC as a function of time using a packaged µSCM shows the sensor data can be used to identify different operation modes: an idle mode and different running programs (MATLAB, Altium Designer, ANSYS Maxwell, etc.).

**Table 1 micromachines-13-01342-t001:** The summary of the reported works for several µSCMs, their contributions, and design methods.

Ref.	Contribution	Design Methodology
[5]	Two planar spiral inductors for crack detection. The sensor output signal is independent of temperature (30~80 °C).	Numerical analysis
[6]	The sensor’s inductor for measurement of the partial discharge of high-voltage equipment.	Not reported
[7]	The induced voltage on the spiral inductor with Fe-Co-B core is utilized to generate electrostatic force to alter the variable capacitor.	FEM models
[15,16,17]	Ultra-low-power (<1 µW) vibrating coils are used to detect DC magnetic field.	Not reported
[22]	Three-axis µSCM is realized by CMOS post-processing.	Theoretical analysis without considering the noise of amplifier and optimization of the sensor layout.
[23]	The flip-chip bonding method for the two-layer planar inductor.	FEM models
This work	General guidelines, novel 4D nomogram, and optimization of µSCM to achieve key performance indices (sensitivity, *SNR*, etc.)	Theoretical analysis considering noise sources of sensor and amplifier; dimensional analysis; scaling analysis

**Table 2 micromachines-13-01342-t002:** Scaling analysis of µSCM and a summary of the key parameters’ exponents for sensitivity (*S_e_*), *e_nR_*, and *SNR_n_* (*T*: temperature; *ρ*: material resistivity; *h*: track thickness).

$S_{e}\propto D_{o}{}^{3}$		$e_{nR}\propto D_{o}{}^{1}$		$SNR_{n}\propto D_{o}{}^{2}$
$SNR_{n}\propto D_{o}{}^{a1}BW^{a2}T^{a3}\rho^{a4}h^{a5}$				
** *a* _1_ **	** *a* _2_ **	** *a* _3_ **	** *a* _4_ **	** *a* _5_ **
2	−0.5	−0.5	−0.5	−0.5

**Table 3 micromachines-13-01342-t003:** A comparison between the fabricated µSCM and commercial magnetic field sensors at 50 Hz input magnetic field and 800 Hz measurement bandwidth.

Ref.	Sensitivity (mV/mT)	Measurement Range (mT)	*SNR* (dB) at 1mT	Type
TI ^1^	200	20	48.69	Hall
AM ^2^	40	±37.5	43.73	Hall
ML ^3^	280	±10	49.03	Hall
AKM ^4^	130	±11	44.7	Hall
[5]	0.005	-	-	µSCM
[6] ^5^	60.9	-	-	µSCM
[22]	6.5	-	-	µSCM
[24] ^6^	1248	-	-	µSCM
This work ^7^	1612	±10	63.5	µSCM

^1^ TI-DRV5056A1/Z1 (Texas Instruments, Dallas, TX, USA); ^2^ A1304 (Allegro microsystem, Manchester, NH, USA); ^3^ MLX91205 (Melexis, Ypres, Belgium); ^4^ EQ-730L (Asahi Kasei Microdevices Corporation, Tokyo, Japan); ^5^ the sensitivity was calculated with the amplifier gain = 1000; ^6^ the sensitivity was calculated with the amplifier gain = 1000 and *D_o_* = 20.7 mm; ^7^ *D_o_* = 5 mm.

## Data Availability

The supporting data are available within the article.

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
