# Peer review of "A Novel Design Nomogram for Optimization of Micro Search Coil Magnetometer for Energy Monitoring in Smart Buildings"

_micromachines, 2022, doi:10.3390/mi13081342_

Round 1

Reviewer 1 Report

The paper is well organized and the content is interesting from the view of sensor applications.
So, the reviewer thinks it can be accepted after several revisions as followings:
1. Please add discussion about detectable field intensity in your results. It can be calculated using noise level and sensitivity.
2. Equation (14) includes, parameter t, the reviewer guesses it use as track thickness, however, the authors it defined “h”. Is it right? If so, please amend this part. Maybe, the authors use “t” as time in other equations.
3.The reviewer thinks the thickness of coil affects the sensor noise. However, the authors avoid to discuss this part, just they commented "The resistivity and thickness of the metal layers are usually predetermined in the fabrication process...”. If there are some explanetions about this point, it is appreciated. Larger thickness induce lower sensrr noise, the reviewer thinks.  
The reviewer wants to know the value of h for calculation of SNR. SNR* depends on Do, Di, s, w, and the authors tried to optimal point for SNR*, it is OK. However, SNR, needs the value of h, the reviewer failed to find the value in the paper.
4. As to measurement, please show the how the authors measured the coil voltage in detail.

Reviewer 2 Report

Dear authors, thank you for submitting your manuscript to Micromachines. It is very strong that you compare design and verification in one manuscript. Your demo of PC power consumption is really cool. The simplicity of the coil manufacturing and electronics will encourage others to reproduce your results.

My main suggestion would be to make a better comparison with the competing commercial sensors. SNR will improve if you make the sensors bigger and pump in more power. You sensor may be bigger, but consumes less power. It would be very strong if you can show that in a table. You should stress that comparison in your final conclusion.

Suggestions from improvement can be found in notes in the attached pdf
